# Diffusion Posterior Proximal Sampling for Image Restoration

Hongjie Wu[*][†]
wuhongjie0818@gmail.com
College of Computer Science, Sichuan
University
Chengdu, China

Linchao He[‡][†]
hlc@stu.scu.edu.cn
National Key Laboratory of
Fundamental Science on Synthetic
Vision, Sichuan University
Chengdu, China

Mingqin Zhang[*]
zhangmingqin@stu.scu.edu.cn
College of Computer Science, Sichuan
University
Chengdu, China

Dongdong Chen
d.chen@hw.ac.uk
Heriot-Watt University
Edinburgh, United Kingdom

Kunming Luo
kluoad@connect.ust.hk
Hong Kong University of Science and
Technology
Hong Kong, China

Mengting Luo[‡]
National Key Laboratory of
Fundamental Science on Synthetic
Vision, Sichuan University
Chengdu, China

Ji-Zhe Zhou[*]
jzzhou@scu.edu.cn
College of Computer Science, Sichuan
University
Chengdu, China

Hu Chen
huchen@scu.edu.cn
College of Computer Science, Sichuan
University
Chengdu, China

Jiancheng Lv[*][§]
lvjiancheng@scu.edu.cn
College of Computer Science, Sichuan
University
Chengdu, China

## Abstract

Diffusion models have demonstrated remarkable efficacy in generating high-quality samples. Existing diffusion-based image restoration algorithms exploit pre-trained diffusion models to leverage data priors, yet they still preserve elements inherited from the unconditional generation paradigm. These strategies initiate the denoising process with pure white noise and incorporate random noise at each generative step, leading to over-smoothed results. In this paper, we present a refined paradigm for diffusion-based image restoration. Specifically, we opt for a sample consistent with the measurement identity at each generative step, exploiting the sampling selection as an avenue for output stability and enhancement. The number of candidate samples used for selection is adaptively determined based on the signal-to-noise ratio of the timestep. Additionally, we start the restoration process with an initialization combined with the measurement signal, providing supplementary information to better align the generative process. Extensive experimental results and analyses validate that our proposed method significantly enhances image restoration performance while consuming negligible additional computational resources.

[*]Also with Engineering Research Center of Machine Learning and Industry Intelligence, Ministry of Education, China.
[†]Equal contribution.
[‡]Also with College of Computer Science, Sichuan University.
[§]Corresponding author.

## CCS Concepts

• **Computing methodologies** → **Computer vision problems**; *Reconstruction*; Machine learning.

## Keywords

Diffusion Model

**ACM Reference Format:**
Hongjie Wu, Linchao He, Mingqin Zhang, Dongdong Chen, Kunming Luo, Mengting Luo, Ji-Zhe Zhou, Hu Chen, and Jiancheng Lv. 2024. Diffusion Posterior Proximal Sampling for Image Restoration. In *Proceedings of the 32nd ACM International Conference on Multimedia (MM '24), October 28-November 1, 2024, Melbourne, VIC, Australia.* ACM, New York, NY, USA, 10 pages. https://doi.org/10.1145/3664647.3681556

## 1 Introduction

Diffusion models [19, 50, 53] have gained broad attention due to the powerful ability to model complex data distribution, and have been applied to a wide range of tasks such as image generation [13, 33, 37], molecule generation [12], and natural language processing [30]. Recent research has demonstrated that pre-trained unconditional diffusion models can be effectively employed to address image restoration problems [9, 26, 47, 49, 54] to leverage rich priors in a plug-and-play fashion, and achieve significant advancements.

However, for diffusion-based image restoration algorithms, they still retain a process inherited from unconditional generation. In particular, (i) these methods start generating the image that corresponds to the measurement with a white noise as initialization, and (ii) a fully stochastic noise is introduced at each step of the generation process. We argue that this paradigm is inappropriate for solving image restoration tasks. Firstly, randomness enriches the diversity of the generated samples in unconditional generation [31, 56]. However, for image restoration tasks where the identity information of the measurement needs to be preserved [57], randomness

DPS      Ours ($n$=2)     Ours ($n$=100)     Reference

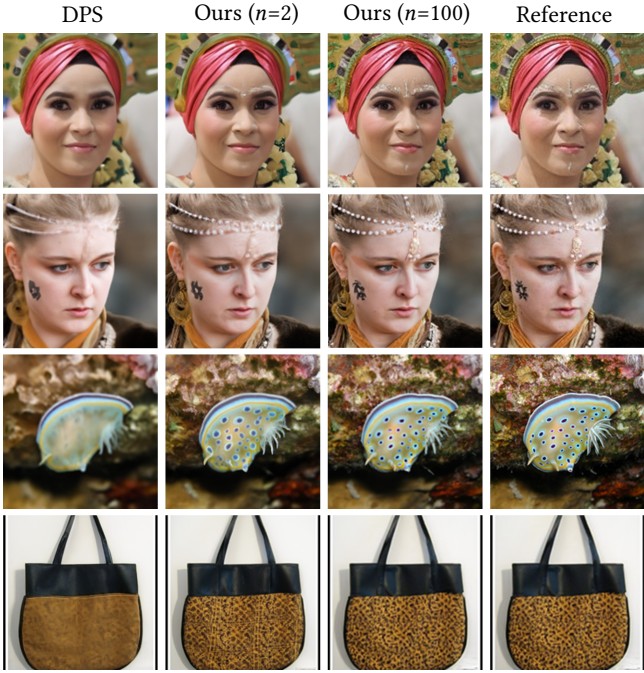

**Figure 1: Examples of the super-resolution ($\times$4) task to illustrate the efficiency of our method, where $n$ denotes the number of candidate samples, and DPS also refers to the case where $n = 1$.**

leads to uncontrollable generation outcomes [4]. Secondly, in the generative process of diffusion-based model, the current state is randomly sampled [8, 47] from the predicted distribution without any correction. Considering the Markovian reverse process, the sampling quality of the next state heavily depends on the current sampling result. If the current result deviates considerably from expectations, the subsequent generative process will encounter a significant exposure bias problem [29, 35]. As a consequence, randomness introduces fluctuations and instability into the generative process, ultimately leading to over-smoothed results.

In this paper, we advance toward a more refined paradigm for solving image inverse problems with pre-trained diffusion models. Specifically, to ensure the sampling result is closely consistent with the measurement identity, we opt for the proximal sample at each step from multiple candidate samples, as opposed to making a random choice. By mitigating uncertainties, the sample consistently progresses toward the desired targets, resulting in a stable generative process and improved output quality. Moreover, we start the generation process with an initialization composed of both the measurement signal and white noise, rather than pure white noise. Consequently, the denoising process commences from a closer starting point, facilitating a faster convergence toward the desired result.

We theoretically analyze that our approach reduces variance in comparison to existing state-of-the-art methods which introduce random sampling. And we conduct extensive experiments to demonstrate the superior restoration capabilities of our method across diverse image restoration tasks, such as super-resolution

(SR), deblurring, and inpainting, with only a marginal additional cost. Furthermore, sufficient experimental analyses are shown to validate the effectiveness of our proposed strategy.

In summary, our contributions are as follows:

- We pioneer to improve the generation quality by exploiting sampling choice from the predicted distribution of each reverse step.
- We propose an efficient proximal sampling strategy that aligns with measurement identity to solve diffusion-based image restoration problems.
- Extensive experimental results demonstrate that our proposed method outperforms other state-of-the-art methods in image restoration performance, requiring only minimal additional computation.

## 2 Background

### 2.1 Denoising Diffusion Probabilistic Models

Denoising Diffusion Probabilistic Models (DDPMs) [19] is a class of models that incorporate a forward (diffusion) process and a corresponding reverse (generative) process. Consider a $T$-step forward process, the noised sample $\mathbf{x}_t$ at time step $t$ can be modeled from the previous state $\mathbf{x}_{t-1}$:

$$q(\mathbf{x}_t|\mathbf{x}_{t-1}) = \mathcal{N}(\mathbf{x}_t; \sqrt{1 - \beta_t}\mathbf{x}_{t-1}, \beta_t \mathbf{I}), \tag{1}$$

where $\mathcal{N}$ denotes the Gaussian distribution, and $\beta_t$ is a pre-defined parameter increasing with $t$. Through reparameterization, $\mathbf{x}_t$ satisfies the following conditional distribution $q(\mathbf{x}_t|\mathbf{x}_0) = \mathcal{N}(\mathbf{x}_t; \sqrt{\bar{\alpha}_t}\mathbf{x}_0, (1 - \bar{\alpha}_t)\mathbf{I})$, where $\bar{\alpha}_t = \prod_{i=0}^{t} \alpha_i$ and $\alpha_t = 1 - \beta_t$.

On the other hand, the reverse process of DDPM is formulated by the transition kernel parameterized by $\theta$:

$$p_\theta(\mathbf{x}_{t-1}|\mathbf{x}_t) = \mathcal{N}(\mathbf{x}_{t-1}; \boldsymbol{\mu}_\theta(\mathbf{x}_t, t), \sigma_t^2 \mathbf{I}). \tag{2}$$

where the learning objective aims to minimize the KL divergence between the forward and backward processes. The epsilon-matching objective is typically set as:

$$\mathbf{L}(\theta) = \mathbb{E}_{\mathbf{x}_0 \sim q(\mathbf{x}_0), \boldsymbol{\epsilon} \sim \mathcal{N}(\mathbf{0}, \mathbf{I}), t \sim \mathbb{U}(\{1, ..., T\})} [\|\boldsymbol{\epsilon} - \boldsymbol{\epsilon}_\theta(\mathbf{x}_t, t)\|^2]. \tag{3}$$

where $\mathbf{x}_0$ is sampled from training data and $\mathbf{x}_t \sim q(\mathbf{x}_t|\mathbf{x}_0)$. Once we have access to the well-trained $\boldsymbol{\epsilon}_\theta(\mathbf{x}_t, t)$, a clean sample can be derived by evaluating the generative reverse process Eq. (2) step by step.

Furthermore, one can view the DDPM equivalent to the variance preserving (VP) form of the stochastic differential equation (SDE) [50]. Accordingly, the epsilon-matching objective Eq. (3) is equivalent to the denoising score-matching [44] objective with different parameterization:

$$\mathbf{L}(\theta) = \mathbb{E}_{\mathbf{x}_0, \boldsymbol{\epsilon}, t} [\|\mathbf{s}_\theta(\mathbf{x}_t, t) - \nabla_{\mathbf{x}_t} \log q(\mathbf{x}_t|\mathbf{x}_0)\|^2]. \tag{4}$$

### 2.2 Diffusion Based Solvers for Image Restoration

This paper focuses on solving the image restoration, or image inverse problems with unconditional diffusion model [8, 26, 47]. Our goal is to retrieve the unknown $\mathbf{x}_0$ from a degraded measurement $\mathbf{y}$:

$$\mathbf{y} = \mathcal{A}(\mathbf{x}_0) + \mathbf{n}, \quad \mathbf{y}, \mathbf{n} \in \mathbb{R}^n, \mathbf{x} \in \mathbb{R}^d, \tag{5}$$

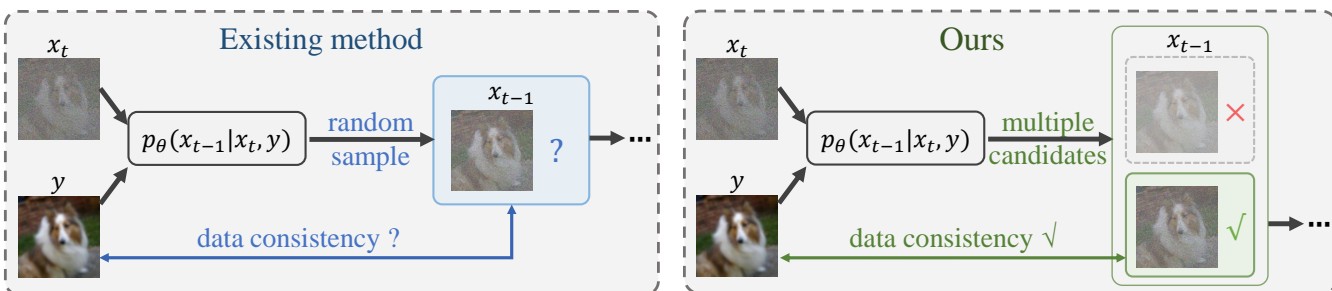

**Figure 2: High-level illustration of our proposed DPPS ($n$=2). Existing methods employ random sampling from the predicted distribution, our approach takes multiple samples and selecting the one with better data consistency (denoted by ✓) at each step.**

where $\mathcal{A}(\cdot) : \mathbb{R}^d \mapsto \mathbb{R}^n$ is the degradation operator and $\mathbf{n} \sim \mathcal{N}(\mathbf{0}, \sigma_y^2 \mathbf{I})$ denotes the measurement noise. The restoration problem can be addressed via conditional diffusion models by substituting the score function $\nabla_{\mathbf{x}_t} \log(\mathbf{x}_t)$ in the reverse-time SDE with the conditional score function $\nabla_{\mathbf{x}_t} \log(\mathbf{x}_t|\mathbf{y})$, which can be derived by Bayes' rule:

$$\nabla_{\mathbf{x}_t} \log p_t(\mathbf{x}_t|\mathbf{y}) = \nabla_{\mathbf{x}_t} \log p_t(\mathbf{x}_t) + \nabla_{\mathbf{x}_t} \log p_t(\mathbf{y}|\mathbf{x}_t). \quad (6)$$

The first prior term can be approximated by a well-trained score network. However, the analytic solution of the second likelihood term is computationally intractable, since there only exists an explicit connection between $\mathbf{y}$ and $\mathbf{x}_0$. To solve this dilemma, Chung et al. [8] propose diffusion posterior sampling (DPS) to approximate the likelihood using a Laplacian approximation: $\nabla_{\mathbf{x}_t} \log p(\mathbf{y}|\mathbf{x}_t) \simeq \nabla_{\mathbf{x}_t} \log p(\mathbf{y}|\hat{\mathbf{x}}_{0|t})$, where $\hat{\mathbf{x}}_{0|t}$ is the denoised estimate via Tweedie's formula [14, 51]. Consequently, the generative distribution $p_\theta(\mathbf{x}_{t-1}| \mathbf{x}_t, \mathbf{y})$ can be modeled as:

$$p_\theta(\mathbf{x}_{t-1}|\mathbf{x}_t, \mathbf{y}) := \mathcal{N}(\mathbf{x}_t;\ \boldsymbol{\mu}_\theta(\mathbf{x}_t, t, \mathbf{y}),\ \sigma_t^2 \mathbf{I}). \quad (7)$$

The mean of Gaussian distribution $\boldsymbol{\mu}_\theta(\mathbf{x}_t, t, \mathbf{y})$ is obtained by:

$$\boldsymbol{\mu}_\theta(\mathbf{x}_t, t, \mathbf{y}) = \frac{1}{\sqrt{\alpha_t}}\left(\mathbf{x}_t - \frac{\beta_t}{\sqrt{1-\bar{\alpha}_t}}\boldsymbol{\epsilon}_\theta(\mathbf{x}_t, t)\right) \\ - \lambda_t \nabla_{\mathbf{x}_t}||\mathbf{y} - \mathcal{A}\hat{\mathbf{x}}_{0|t}||, \quad (8)$$

where $\hat{\mathbf{x}}_{0|t} = \frac{1}{\sqrt{\bar{\alpha}_t}}\left(\mathbf{x}_t - \sqrt{1-\bar{\alpha}_t}\boldsymbol{\epsilon}_\theta(\mathbf{x}_t, t)\right)$, and $\lambda_t$ is a tunable step size.

## 3 Diffusion Posterior Proximal Sampling

### 3.1 Random Sampling Induces Uncertainties and Error Accumulation

Existing approaches [9, 26, 47, 49, 54] address image restoration or inverse problems following a process derived from unconditional generation. Specifically, the process (i) initiates the denoising process with pure white noise, and (ii) incorporates random noise at each reverse step. We contend that randomness is unsuitable for restoration problems that demand the preservation of measurement identities, such as super-resolution or deblurring. Furthermore, the cumulative impact of random noise at each step results in the smoothing of output, thereby yielding low-quality generated

samples. Recent investigations [16, 32] align with our findings, highlighting that randomness introduces instability and fluctuations, ultimately culminating in suboptimal samples.

On the other hand, we observe a discrepancy in the inputs provided to the noise prediction network $\boldsymbol{\epsilon}_\theta$. During the training phase, the network is fed with ground truth training samples. However, in the inference stage, the input $\mathbf{x}_t$ is randomly sampled from the predicted distribution. In cases where the predicted distribution is inaccurate, random sampling can exacerbate the deviation from the expected values, introducing substantial exposure bias [35] to the generative process. While the data consistency update $\nabla_{\mathbf{x}_t}||\mathbf{y} - \mathcal{A}\hat{\mathbf{x}}_{0|t}||$ does offer mitigation by adjusting the sample to align with the measurement, it demands delicate design and specifically-tuned parameters [48]. And the optimal parameter values vary across datasets and tasks. Consequently, the selection of sampling choices holds significance as it directly influences the input to the network, thereby affecting the generation quality.

### 3.2 Proximal Sampling at Each Step

To tackle the challenges posed by random sampling, in this paper, we propose to extract multiple candidate samples from the predicted distribution, and select the most proximal one [36] to our anticipated target. Our motivation comes from the following idea: considering $\mathbf{x}_0$ is an ideal but unknown solution for the image restoration problem, the sample taken from the predicted distribution $\mathbf{x}_{t-1} \sim p_\theta(\mathbf{x}_{t-1}|\mathbf{x}_t, \mathbf{y})$ should be close to posterior $q(\mathbf{x}_{t-1}|\mathbf{x}_t, \mathbf{x}_0)$, as it models the desired reverse process.

The mathematical formulation of our selection process is given by:

$$\mathbf{x}_{t-1} = \underset{\mathbf{x}_{t-1}^i}{\arg\min} ||\mathbf{x}_{t-1}^i - \mathbf{x}_{t-1}^*||_2^2 \quad (9)$$

where $\mathbf{x}_{t-1}^i \sim p_\theta(\mathbf{x}_{t-1}|\mathbf{x}_t, \mathbf{y})$, $i \in [0, n-1]$, and $\mathbf{x}_{t-1}^*$ is our proposed deterministic solution [1] via DDIM [46] with unknown $\mathbf{x}_0$ (see supplementary material for detailed derivation):

$$\mathbf{x}_{t-1}^* = \sqrt{\bar{\alpha}_{t-1}}\mathbf{x}_0 + \sqrt{1-\bar{\alpha}_{t-1}} \cdot \frac{\mathbf{x}_t - \sqrt{\bar{\alpha}_t}\mathbf{x}_0}{\sqrt{1-\bar{\alpha}_t}} \\ = C_1\mathbf{x}_t + C_2\mathbf{x}_0. \quad (10)$$

---

[1] the symbol * means an ideal but unknown solution.

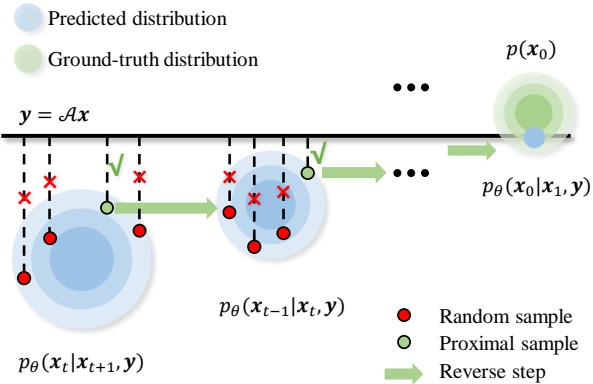

**Figure 3: Conceptual illustration of our DPPS. Our method extract multiple candidate samples from the predicted distribution and choose the one with the highest measurement consistency.**

Here $C_1 = \sqrt{1 - \bar{\alpha}_{t-1}}/\sqrt{1 - \bar{\alpha}_t}$ and $C_2 = \sqrt{\bar{\alpha}_{t-1}} - \sqrt{\bar{\alpha}_t}\sqrt{1 - \bar{\alpha}_{t-1}}/\sqrt{1 - \bar{\alpha}_t}$ are introduced for simplicity.

The underlying motivation of our approach is depicted in Figure 3. Without changing the diffusion denoiser, our approach directs the sampling results toward a predefined target, mitigating the drawbacks induced by randomness. Moreover, the generative process remains stochastic, benefiting from the injected noise [1, 24] since it corrects prediction errors and imprecise parameter settings from previous steps. Consequently, our method converges more rapidly and attains superior results.

However, accessing $\mathbf{x}_0$ during inference is infeasible, rendering $\mathbf{x}_{t-1}^*$ also unknown. Now, one important contribution of this paper is to project Eq. (9) onto the measurement subspace. This is feasible as $\mathcal{A}\mathbf{x}_{t-1}^*$ can be approximated under the condition $\mathbf{y} \approx \mathcal{A}\mathbf{x}_0$ when the degradation operator is linear and $\sigma_y$ is assumed within a moderate range.

$$\mathcal{A}\mathbf{x}_{t-1}^* \approx C_1\mathcal{A}\mathbf{x}_t + C_2\mathbf{y}. \tag{11}$$

Specifically, by projecting onto the measurement subspace, we choose $\mathbf{x}_{t-1}$ by the following:

$$\mathbf{x}_{t-1} = \arg\min_{\mathbf{x}_{t-1}^i} \|\mathcal{A}(\mathbf{x}_{t-1}^i - \mathbf{x}_{t-1}^*)\|_2^2 \tag{12}$$

Since $\mathbf{x}_0$ is expected to be more accurate than the denoised result $\hat{\mathbf{x}}_{0|t}$, the measurement $\mathbf{y} \approx \mathcal{A}\mathbf{x}_0$ can provide a stable and strong supervisory signal to correct the sampling result. Then by means of sample selection, we effectively control the result $\mathbf{x}_{t-1}$ within a more desirable region, even though we have no access to the $\mathbf{x}_0$.

As $\mathcal{A}$ is irreversible and complex in numerous scenarios [6, 7], obtaining its pseudo-inverse poses challenges. Therefore, we propose to randomly sample $n$ candidates $\mathbf{x}_{t-1}^i, i \in [0, n-1]$ from the predicted distribution $p_\theta(\mathbf{x}_{t-1}|\mathbf{x}_t, \mathbf{y})$, and choose the one with the minimal deviation from our anticipated values. Finally, Eq. (9) becomes:

$$\mathbf{z}_t = \arg\min_{\mathbf{z}_t^i} \|\mathcal{A}\left(\boldsymbol{\mu}_\theta(\mathbf{x}_t, t, \mathbf{y}) + \sigma_t\mathbf{z}_t^i - C_1\mathbf{x}_t\right) - C_2\mathbf{y}\|_2^2. \tag{13}$$

The process of sample selection is also denoted as noise selection, given by $\mathbf{x}_{t-1}^i = \boldsymbol{\mu}_\theta(\mathbf{x}_t, t, \mathbf{y}) + \sigma_t\mathbf{z}_t^i$. We detail the full algorithm in

---

**Algorithm 1** Diffusion Posterior Proximal Sampling

**Require:** $T, \mathbf{y}, \{\lambda_t\}_{t=1}^T, \{\sigma_t\}_{t=1}^T, n$
1: $\mathbf{x}_T = \sqrt{\bar{\alpha}_T}\mathcal{A}^T\mathbf{y} + \sqrt{1 - \bar{\alpha}_T}\boldsymbol{\epsilon}, \quad \boldsymbol{\epsilon} \sim \mathcal{N}(\mathbf{0}, \mathbf{I})$
2: **for** $t = T$ **to** 1 **do**
3:     compute $\boldsymbol{\mu}_\theta(\mathbf{x}_t, t, \mathbf{y})$ via (8)
4:     **for** $i = n$ **to** 1 **do**
5:         $\mathbf{z}_t^i \sim \mathcal{N}(\mathbf{0}, \mathbf{I})$
6:         $D_i = \|\mathcal{A}(\boldsymbol{\mu}_\theta(\mathbf{x}_t, t, \mathbf{y}) + \sigma_t\mathbf{z}_t^i - C_1\mathbf{x}_t) - C_2\mathbf{y}\|_2^2$
7:     **end for**
8:     $\mathbf{z}_t = \arg\min_{\mathbf{z}_t^i} D_i$
9:     $\mathbf{x}_{t-1} \leftarrow \boldsymbol{\mu}_\theta(\mathbf{x}_t, t, \mathbf{y}) + \sigma_t\mathbf{z}_t$
10: **end for**
11: **return** $\hat{\mathbf{x}}_0$

---

Algorithm 1, and name our algorithm Diffusion Posterior Proximal Sampling (DPPS). The chosen $\mathbf{x}_{t-1}$ is referred to as the proximal sample due to the similarity to the proximal operator [36].

### 3.3 Adaptive Sampling Frequency

It is commonly perceived that the generative process of diffusion does not exhibit uniform significance across all timesteps [15, 24, 27]. Based on this insight, we adaptively adjust the number of candidate samples during the generative process according to the signal-to-noise ratio to enhance image quality and reduce computational cost. Through extensive experimentation, we discovered that using a larger number of candidate samples in the final stage of generation yields better results. The number of candidate samples $n$ at each step can be expressed as:

$$n = \max(\lfloor N_{max} * (1 - e^{-\lambda_t})\rfloor, 2), \tag{14}$$

where $t \in [0, 999]$ is the timestep, $\lambda_t = \frac{\bar{\alpha}_t}{1 - \bar{\alpha}_t}$ represents the signal-to-noise ratio, and $N_{max}$ is a hyper-parameter that adjusts the maximum sampling frequency. The minimum frequency is set to 2, ensuring our proximal sampling strategy.

### 3.4 Aligned Initialization

Recent studies have pointed out that initializition have a significant impact on the generated results [4, 32, 43], and the discrepancy between the two distributions account for a discretization error [2]. In this paper, inspired by [16], we simply initialize the sample in the same way as during training, making the best use of the available measurement

$$\mathbf{x}_T = \sqrt{\bar{\alpha}_T}\mathcal{A}^T\mathbf{y} + \sqrt{1 - \bar{\alpha}_T}\boldsymbol{\epsilon} \quad \boldsymbol{\epsilon} \sim \mathcal{N}(\mathbf{0}, \mathbf{I}). \tag{15}$$

where $\mathcal{A}^T$ means the transpose of operator. We argue this trick provides a modicum of information as a signal to the reverse model, as it realigns the distribution of initial latent with the training distribution [16].

### 3.5 Discussion

*3.5.1 Relevance to Existing Methods.* DPS [8] proposes to estimate the intractable likelihood under Laplacian approximation, and adopts random sampling from the predicted distribution. However, the randomly chosen sample may not be well-coordinated

| Input | DDRM | DPS | DiffPIR | Ours | Reference |
|---|---|---|---|---|---|

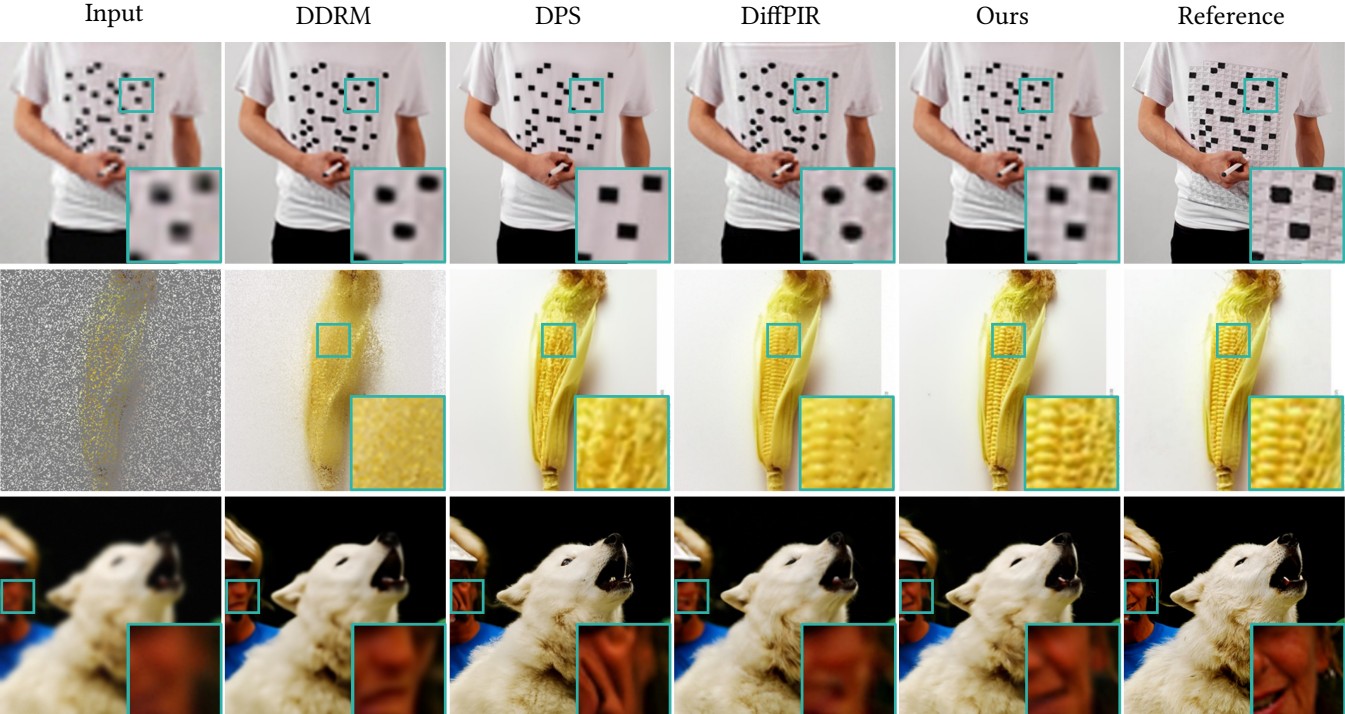

**Figure 4: Image restoration results with $\sigma_y = 0.01$. Row 1: SR (×4), Row 2: 80% inpainting, Row 3: Gaussian deblurring.**

with the measurement information. Another study MCG [9] iteratively applies projection onto the measurement subspace after the denoising step to ensure data consistency. However, the imposed projections lead to accumulated error due to measurement noise.

Our algorithm can be viewed as a special case of DPS, where we choose the sample that exhibits better measurement consistency. It absorbs the advantages of the robustness to noise from DPS and the faithful data consistency of MCG. Moreover, the better data consistency is achieved by injecting a guided adaption $z_t$, which also helps correct the prediction errors in last generation steps [23, 24], leading to enhanced output quality.

*3.5.2 Computational Efficiency.* Selecting the proximal sample introduces some extra computational overhead. In practice, the diffusion model conducts back-propagation only once, aligning with the mainstream approaches [8, 9, 48, 49]. The evaluations for Eq. (13) are considerably cheaper than gradient calculation, making the additional computational costs manageable. We show the computational resources for different settings in Section 5.4.

*3.5.3 Theoretical Analysis.* Here, we provide some theoretical supports for our methodology. Despite our proposed proximal sampling being random, we can theoretically approach the desired point within an upper bound with mild assumptions. The proposition, detailed in supplementary material, proves that we can converge to our desired target $x^*_{t-1}$, when the number of candidate samples is large enough. Furthermore, our method doesn't require an exceptionally large number of candidates to achieve promising performance. This is because the variance of our method decreases when the proximal one is selected from any number of candidates.

PROPOSITION 3.1. *For the random variable* $z \sim \mathcal{N}(0, I)$ *and its objective function:*

$$f(z) = \|\mathcal{A}(\mu_\theta(x_t, t, y) + \sigma_t z - C_1 x_t) - C_2 y\|_2^2. \quad (16)$$

*The variance for $f(z)$ is denoted by $Var(f(z))$. We have*

$$Var_{DPS}(f(z)) > Var_{MC}(f(z)) > Var_{Ours}(f(z)). \quad (17)$$

*Here, $Var_{DPS}(f(z))$ is the variance of DPS [8], $Var_{MC}(f(z))$ is the variance of Monte Carlo sampling, and $Var_{Ours}(f(z))$ is the variance of our proximal sampling method.*

In line with our theoretical analysis, empirical observations suggest that the proximal sample strategy demonstrates improved data consistency and reduced stochasticity, as evidenced by the experimental results.

## 4 Related Work

### 4.1 Diffusion Models for Image Restoration

To solve image restoration or image inverse problems with diffusion models, plenty of notable works [8, 13, 26, 47] have been introduced. The first category of approaches [13, 41, 42] involves training conditional diffusion models or approximating the likelihood with synthetic image pairs as training data. However, these methods require specific training for each task and lack generalization across a wide range of inverse problems. In contrast, the second category of approaches addresses inverse problems with unconditional diffusion models by guiding the reverse diffusion process [49, 59]. Several studies [8, 47, 48] concentrate on estimating the time-dependent likelihood $p_t(y|x_t)$ for posterior sampling.

**Table 1: Quantitative evaluation of image restoration tasks on FFHQ 256×256-1k and ImageNet 256×256-1k with $\sigma_y = 0.01$. Bold: best, underline: second best.**

| Method | Inpaint (random) | | | Deblur (Gaussian) | | | Deblur (motion) | | | SR (× 4) | | |
|---|---|---|---|---|---|---|---|---|---|---|---|---|
| | PSNR ↑ | LPIPS ↓ | FID ↓ | PSNR ↑ | LPIPS ↓ | FID ↓ | PSNR ↑ | LPIPS ↓ | FID ↓ | PSNR ↑ | LPIPS ↓ | FID ↓ |
| FFHQ | | | | | | | | | | | | |
| PnP-ADMM [5] | 27.32 | 0.349 | 63.19 | 25.97 | 0.260 | 94.50 | 25.86 | 0.278 | 59.06 | 26.94 | 0.292 | 90.11 |
| Score-SDE [50] | 21.91 | 0.371 | 58.19 | 18.37 | 0.629 | 169.68 | 15.79 | 0.635 | 176.73 | 24.10 | 0.363 | 75.50 |
| MCG [10] | 24.59 | 0.265 | 29.31 | 19.67 | 0.497 | 85.85 | 18.00 | 0.604 | 91.73 | 24.02 | 0.321 | 52.88 |
| DDRM [26] | 21.85 | 0.378 | 78.40 | 26.38 | 0.298 | 62.72 | - | - | - | **27.31** | 0.248 | 52.49 |
| DPS [8] | 27.29 | 0.182 | 23.11 | 26.04 | 0.228 | 30.52 | 25.24 | 0.261 | 33.47 | 26.55 | 0.237 | 34.50 |
| LGD-MC [48] | 26.11 | 0.257 | 35.26 | 24.08 | 0.288 | 34.54 | 21.04 | 0.370 | 49.19 | 26.12 | 0.247 | 33.68 |
| DiffPIR [59] | **27.96** | 0.210 | 30.38 | 25.38 | 0.276 | 32.00 | 22.74 | 0.331 | 83.42 | 24.74 | 0.273 | 33.03 |
| Ours | 27.50 | **0.161** | **18.65** | 26.42 | **0.221** | **28.93** | 27.82 | **0.197** | 28.28 | 26.94 | **0.201** | **25.98** |
| ImageNet | | | | | | | | | | | | |
| PnP-ADMM [5] | 25.14 | 0.405 | 66.54 | 21.97 | 0.419 | 63.15 | **21.86** | 0.408 | 61.46 | 23.95 | 0.346 | 71.41 |
| Score-SDE[50] | 16.25 | 0.653 | 102.56 | 21.31 | 0.467 | 82.54 | 13.56 | 0.661 | 89.62 | 17.69 | 0.624 | 96.67 |
| MCG [10] | 23.21 | 0.324 | 44.09 | 12.31 | 0.647 | 109.45 | 18.32 | 0.633 | 99.26 | 17.08 | 0.538 | 85.91 |
| DDRM [26] | 19.34 | 0.555 | 147.00 | **23.67** | 0.401 | 66.99 | - | - | - | **25.49** | 0.319 | 54.77 |
| DPS [8] | 25.65 | 0.240 | 29.04 | 19.65 | 0.422 | 65.35 | 18.79 | 0.458 | 77.29 | 23.88 | 0.335 | 42.83 |
| LGD-MC [48] | 24.06 | 0.316 | 40.95 | 20.32 | 0.423 | 62.79 | 19.07 | 0.461 | 78.79 | 22.78 | 0.390 | 59.61 |
| DiffPIR [59] | **25.85** | 0.235 | 33.16 | 22.03 | 0.395 | 54.71 | 19.86 | 0.433 | 79.23 | 24.78 | 0.302 | 39.25 |
| Ours | 24.97 | **0.217** | 24.90 | 22.70 | **0.364** | 51.21 | 21.65 | **0.375** | 51.35 | 24.44 | **0.267** | **30.70** |

Among them, Song et al. [48] enhances the likelihood approximation with a Monte-Carlo estimate. Ma et al. [32] tackles image super-resolution by selecting the best starting point. [20–22] effectively constrain the results of each generative step, adeptly tackling low-light image enhancement and shadow removal.

Recently, latent diffusion models have also seen advancements [37, 52] and have been widely adopted in various image inverse problem scenarios [11, 17, 38, 39, 45]. Furthermore, while several deep learning-based proximal optimization algorithms have been proposed [28, 55], our method stands out as the first to make the diffusion sampling as a proximal optimization process.

## 4.2 The Exposure Bias in Diffusion Models

The exposure bias [29, 35] in diffusion models is described as the misalignment between the training input $\mathbf{x}_t$ and the reverse process input $\mathbf{x}_t$, which is essentially a mismatch between the predicted noise $\epsilon_\theta$ and the actual noise $\epsilon$. Ning et al. [35] mitigate the exposure bias problem by perturbing the training input, rendering the network more robust to inaccurate inputs during generation. Ning et al. [34] propose dynamic scaling to correct the magnitude error of $\epsilon_\theta$ and improve sample quality. Li et al. [29] present a novel solution by adjusting the corruption level of the current samples. This paper provides a new way to effectively alleviate the exposure bias problem to some extent by reducing the uncertainty of random sampling.

## 5 Experiments

### 5.1 Setup

*Dataset, Model.* To showcase the effectiveness of the proposed methods, we conducted experiments on two standard datasets,

namely FFHQ 256×256 [25] and ImageNet 256×256 [40]. The evaluation encompasses the first 1k images in the validation set of each dataset. The diffusion models pre-trained on ImageNet and FFHQ are sourced from Dhariwal and Nichol [13] and Chung et al. [8], respectively. For comprehensive comparisons, we include state-of-the-art diffusion-based image restoration solvers, including the Plug-and-play alternating direction method of multipliers (PnP-ADMM) [5], Score-SDE [49], Denoising Diffusion Restoration Models (DDRM) [26], Manifold Constrained Gradient (MCG) [9], DPS [8], LGD-MC [48], and DiffPIR [59]. Sampling frequency parameter $N_{max}$ is set to 50. To ensure fair comparisons, we utilized the same pre-trained diffusion models for all diffusion-based methods. All experiments were executed with a fixed random seed.

*Degradation Operator.* The degradation operators are defined as follows: (i) For inpainting, 80% of the pixels (all RGB channels) in the image are masked. (ii) For Gaussian blur, a blur kernel of size $61×61$ with a standard deviation of 3.0 is employed. (iii) For motion blur, the kernel, following the procedure outlined in Chung et al. [9], has a size of $61×61$ and an intensity value of 0.5. (iv) For SR × 4, the operator involves 4× bicubic down-sampling. Additive Gaussian noise with a variance of $\sigma_y = 0.01$ is applied for all degradation.

*Metrics.* The metrics employed for the comparison encompass: Peak Signal-to-Noise Ratio (PSNR) as distortion metrics; Learned Perceptual Image Patch Similarity (LPIPS) [58] distance, and Frechet Inception Distance (FID) [18] as perceptual metrics.

### 5.2 Main Results

We present the statistic results of the general image restoration tasks on both FFHQ and ImageNet datasets, as detailed in Table 1. To the dataset with homogeneous scenarios, i.e., FFHQ, our proposed

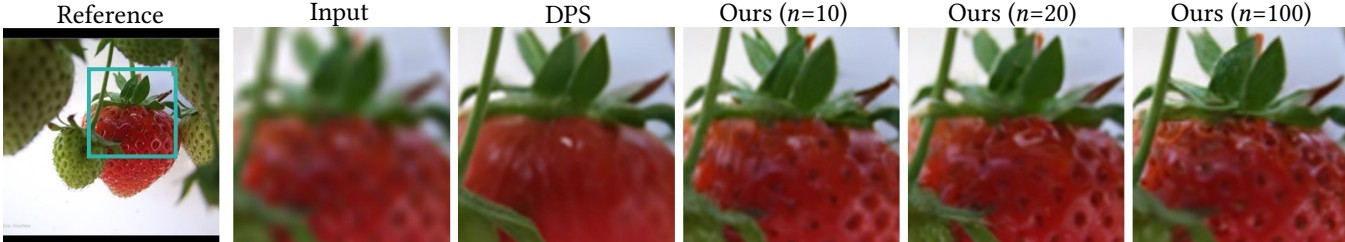

| Reference | Input | DPS | Ours ($n$=10) | Ours ($n$=20) | Ours ($n$=100) |

**Figure 5: Visual results on SR ($\times$4) task to demonstrate the efficacy of our proposed method and to explore the impact of $n$.**

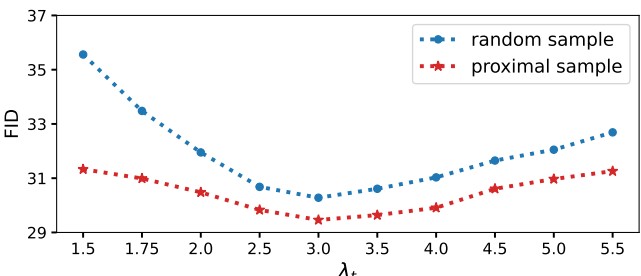

**Figure 6: FID comparison between random sampling and proximal sampling with different values of $\lambda_t$.**

**Table 2: Comparision of computational resources and performance with different $n$ on FFHQ SR$\times$ 4.**

| Setting | Memory / Growth | Speed / Growth | LPIPS / Gain |
|---|---|---|---|
| $n$=1 | 2599 MB / 0.0 % | 56.36 s / 0.0 % | 0.235 / 0.0 % |
| $n$=2 | 2603 MB / 0.2 % | 56.64 s / 0.5 % | 0.221 / 6.0 % |
| $n$=10 | 2621 MB / 0.8 % | 56.82 s / 0.8 % | 0.207 / 11.9 % |
| $n$=20 | 2643 MB / 1.7 % | 57.25 s / 1.5 % | 0.203 / 13.6 % |
| adaptive $n$ | 2714 MB / 4.4 % | 56.86 s / 0.9 % | 0.201 / 14.5% |

method demonstrates impressive results across all metrics, establishing its superiority over existing state-of-the-art methods. When dealing with more varied images within the ImageNet dataset, our approach exhibits substantial outperformance against all baseline methods in terms of FID and LPIPS, while maintaining comparable levels in PSNR.

The visual results for inpainting, SR, and Gaussian deblurring are shown in Figure 4, showcasing the evident superiority of our proposed method. While DDRM attains commendable distortion results with fewer Neural Function Evaluations (NFEs), it faces limitations in reliably restoration results for image inpainting tasks characterized by a very low rank of the measurement. DiffPIR achieves satisfactory results in various scenarios; however, its performance is tied to the analytical solution for the data consistency term and exhibits sensitivity to measurement noise. DPS differs from our approach in that it introduces random noise at each generation step, leading to over-smoothed and unstable restoration results, as depicted in Figure 4 (column 3). Conversely, our proposed method circumvents such drawbacks. The generative process is stabilized by directing the sample to a predefined target, yielding realistic and detailed restoration outcomes. It is noteworthy that the generated results by our method exhibit minimal variation with different seeds (refer to supplementary material), aligning with the intended design of identity preserving.

## 5.3 The Effect of Proximal Sampling

*5.3.1 Faster Convergence.* To investigate the impact of our proximal sampling on the generated results, we conducted further experimental analyses using the first 150 images from the FFHQ validation set. In addition to the naive DPS, we incorporated the results of DPS with DDIM [46] deterministic sampling, denoted as DPS_DDIM, which mitigates the effects of randomness. Results for (a) mean

of $\|\mathbf{y} - \mathcal{A}\hat{\mathbf{x}}_{0|t}\|_2^2$, (b) average LPIPS of $\hat{\mathbf{x}}_{0|t}$, and (c) average PSNR of $\hat{\mathbf{x}}_{0|t}$ over timesteps are reported in Figure 7. It is evident that our method facilitates a more stable optimization process, yielding markedly superior results compared to DPS and DPS_DDIM within the same period. This observation also supports the claim that the measurement can provide reliable supervisory signals for the sampling result.

*5.3.2 More Robust to Hyper-parameter.* We anticipate that our proposed proximal sampling serves as an adaptive correction for prediction errors resulting from imprecise parameter settings. To substantiate this claim, we conducted experiments with different $\lambda_t$, comparing the results of our proximal sampling with those obtained through random sampling. As illustrated in Figure 6, our method consistently outperforms random sampling across different parameter settings. Notably, the performance of the random sampling method exhibits fluctuations with changes in parameters. In contrast, our method demonstrates stability over a range of parameter configurations, thereby validating the assertion.

## 5.4 Computational Resource

To examine the impact of the variable $n$ on computational efficiency, we conducted an evaluation that consisted of measuring memory consumption and computational time required for the SR$\times$4 task. This evaluation was performed on a single NVIDIA RTX 3090 GPU, and the results are presented in Table 2. Notably, when the value of $n$ is relatively small ($\leq$20), our method shows a substantial enhancement in the restoration metric as $n$ increases while incurring minimal increases in memory consumption and computation time. Additionally, the carefully designed adaptive sampling frequency strategy further enhances restoration efficiency within less computation time. These findings highlight the effectiveness and validity of our approach.

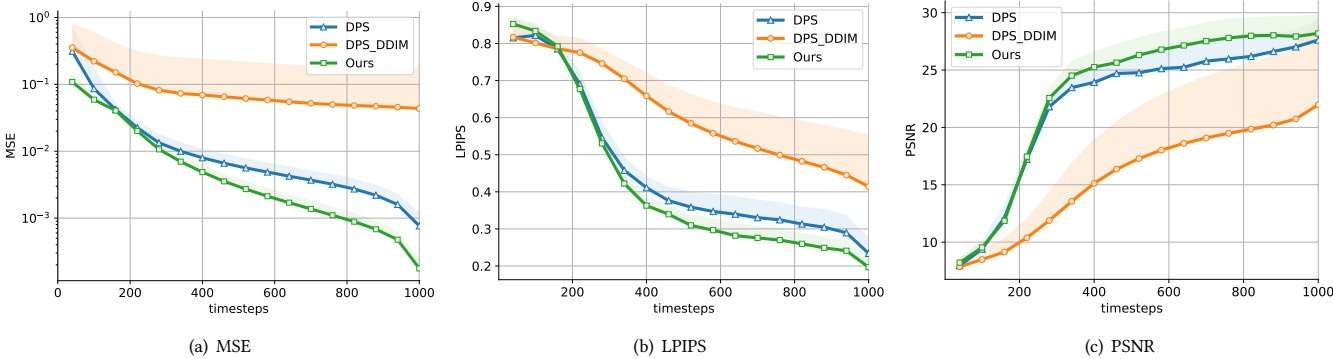

(a) MSE

(b) LPIPS

(c) PSNR

**Figure 7: Convergence speed analysis. (a) mean error $\|\mathbf{y} - \mathcal{A}\hat{\mathbf{x}}_{0|t}\|_2^2$, (b) average LPIPS of $\hat{\mathbf{x}}_{0|t}$, and (c) average PSNR of $\hat{\mathbf{x}}_{0|t}$ with timesteps respectively. Our method achieves a faster optimization process and better restoration performance.**

**Table 3: Ablation studies on the number of candidate samples on FFHQ SR× 4.**

|  | FFHQ | | | ImageNet | | |
|---|---|---|---|---|---|---|
| Method | PSNR↑ | LPIPS↓ | FID↓ | PSNR↑ | LPIPS↓ | FID↓ |
| $n$=1 | 26.44 | 0.235 | 34.91 | 23.95 | 0.336 | 43.54 |
| $n$=2 | 26.64 | 0.221 | 31.19 | 24.42 | 0.328 | 40.63 |
| $n$=10 | 26.77 | 0.207 | 27.60 | 24.64 | 0.282 | 34.05 |
| $n$=20 | 26.93 | 0.203 | 26.30 | **24.79** | 0.276 | 33.75 |
| adaptive $n$ | **26.94** | **0.201** | **25.98** | 24.44 | **0.267** | **30.70** |

**Table 4: Comparison of the impact of different noise levels.**

| $\sigma_y$ | Method | Inpainting | | | SR×4 | | |
|---|---|---|---|---|---|---|---|
|  |  | PSNR↑ | LPIPS↓ | FID↓ | PSNR↑ | LPIPS↓ | FID↓ |
| 0.0 | Random | **27.69** | 0.173 | 21.11 | 26.61 | 0.235 | 34.55 |
|  | Ours | 27.61 | **0.156** | **17.50** | **27.42** | **0.187** | **24.08** |
| 0.01 | Random | 27.29 | 0.182 | 23.11 | 26.55 | 0.237 | 34.50 |
|  | Ours | **27.50** | **0.161** | **18.65** | **26.94** | **0.201** | **25.98** |
| 0.05 | Random | **26.68** | 0.228 | 30.63 | 25.89 | 0.257 | 34.59 |
|  | Ours | 26.51 | **0.208** | **24.62** | **26.07** | **0.245** | **30.67** |

## 5.5 Ablation Study

*5.5.1 The Number of Candidate Samples.* The number of samples, denoted as $n$, stands out as a pivotal parameter of the proposed methodology. The ablation studies relating to $n$ are shown in Table 3 and Figure 5. Figure 5 illustrates a noticeable enhancement in output quality and details with increasing $n$. This observation is reinforced by Table 3, where both the LPIPS and FID metrics exhibit a notable decrease as $n$ increases. The augmented number of candidate samples corresponds to a higher likelihood of proximity to the intended target, thereby providing enhanced supervision for the generative process. The enhancement of the proposed adaptive sampling frequency further enhance the efficiency and applicability of our proposed method.

It is noteworthy that the improvement becomes relatively marginal when $n$ exceeds 20 and may even lead to a decrease in PSNR. There are several reasons for this phenomenon. (i) The approximation is only conducted within the range space, neglecting the null space. This can lead to a certain degree of misalignment with the reference image. (ii) Since we utilized $\mathbf{y} \approx \mathcal{A}\mathbf{x}_0$ in our algorithm, the presence of measurement noise ($\mathbf{n}$ in Eq. (5)) affects the accuracy of our sample selection. (iii) It can also be attributed to the well-known trade-off between perception and distortion metrics [3]. Consequently, a larger value of $n$ does not necessarily lead to better results.

*5.5.2 Noise Level.* The precision of the approximation is impacted by the level of measurement noise $\mathbf{n}$. We explore the effects of

different noise levels on the proposed method compared with random sampling. The results presented in Table 4 reveal that (i) both methods experience a decline in performance as the noise level increases, and our approach consistently outperforms random sampling in all settings. (ii) For inpainting, both methods experience considerable degradations as the noise level increases, demonstrating sensitivity to the noise levels. (iii) For SR, notable performance degradation is observed in our method as the noise level increases, while the method adopting random sampling demonstrates greater robustness in this regard.

## 6 Conclusion

In this paper, we introduce a novel approach to mitigate problems caused by misaligned random sampling in diffusion-based image restoration methods. Specifically, our approach advocates selecting the proximal sample that is more consistent with the observed measurement in the predicted distribution. An adaptive sampling frequency strategy is followed to optimize the computational efficiency of the proposed method. In addition, we propose a realignment of the inference initialization involving measurement information to better align with expected generations. Experimental results validate a substantial performance improvement compared to SOTA methods. Our method innovatively takes advantage of stochastic sampling in the diffusion generative process, exploiting the sampling selection to enhance the generation quality and offering new insights for future diffusion inference algorithms.

# Acknowledgments

This work was supported by the Fundamental Research Funds for the Central Universities (No. 1082204112364), the National Nature Science Foundation of China (No. 62106161), and the Sichuan University Luzhou Municipal Government Strategic Cooperation Project (No. 2022CDLZ-8). We thank the anonymous reviewers for their insightful comments and feedback, and extend our gratitude to Mengxi Xie from SJTU and Linrui Dai from CQU for their constructive suggestions on the figures.

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
