# OpenReview forum: "Diffusion Posterior Proximal Sampling for Image Restoration"
_acmmm.org/ACMMM/2024/Conference — MM2024 Oral_

### Official Review · Reviewer_opvt · 2024-05-24

**Rating:** 4
**Confidence:** 3

**Summary:**

The paper introduces an approach called Diffusion Posterior Proximal Sampling (DPPS) for image restoration using diffusion models. It enhances the restoration process by selecting samples that are more consistent with the measurement identity at each generation step and by initializing with a combination of the measurement signal, which significantly improves performance with minimal additional computational cost.

**Strengths:**

1. The DPPS method improves the stability of the generative process by reducing randomness, resulting in higher quality image restoration outcomes.
2. The research provides empirical evidence of the superior performance of the proposed method across various image restoration tasks such as super-resolution, deblurring, and inpainting.
3. The initialization strategy and proximal sampling strategy presented in the paper offer new perspectives and directions for the application of diffusion models in image restoration.

**Limitations:**

1. The article seems to use the results of each step to constrain the trajectory. There have been many previous application articles [1] [2] [3] that have considered similar research. The author should compare and analyze them to further confirm the innovation.

[1] Global structure-aware diffusion process for low-light image enhancement

[2] Low-light image enhancement with wavelet-based diffusion models

[3] DeS3: Adaptive Attention-Driven Self and Soft Shadow Removal Using ViT Similarity

2. For the experimental part, especially the visual comparison, the authors should provide a more detailed analysis of why more details are lost as the number of n increases.
3. The author should provide more details on the computational complexity and parameter size of the overall architecture compared to previous methods.
4. Can the proposed method be extended to other tasks? If so, it will be very helpful.

**Suitability:**

2

---

### Official Review · Reviewer_kajr · 2024-05-24

**Rating:** 3
**Confidence:** 2

**Summary:**

The paper presents a pioneering approach to enhance the quality of image generation by strategically utilizing sampling choices from the predicted distribution at each reverse step in the diffusion process. It introduces an efficient proximal sampling strategy that is closely aligned with the measurement identity, which is particularly effective for solving diffusion-based image restoration problems. The extensive experimental results provided in the paper demonstrate that this proposed method surpasses other state-of-the-art techniques in terms of image restoration performance, with the added benefit of requiring only minimal additional computation.

**Strengths:**

1. The methods proposed in this paper appear to be highly effective in achieving data consistency in diffusion-based image restoration methods, which I find to be quite challenging while this paper manages to accomplish it.
2. Surprisingly, the solutions presented in this paper only require a single time model inference, significantly enhancing the practicality of the proximal sampling strategy. In fact, I have read a similar work requiring multiple sampling. The specific implementation in this paper exceeds my expectations, and I consider it to be interesting and novel.
3. The image restoration results presented in this paper, especially on data consistency, are impressive.

**Limitations:**

1. I find the handling of noise in the degradation model in this paper somewhat confusing. Eq. (5) suggests that $y=A(x_0)+n$, yet in L350, it is stated that $y≈Ax^*_0$. I wonder why the noise $n$ can be ignored? The method maybe fails when noise is large? (e.g., $\sigma_y=50/255$)
2. For the degration process $A(x)$, I think it can not always be equivalent to $Ax$, which means multiplicative distribution law can not be casually used in Eq.(11)~(13). (maybe need an additional statement)
3. $x_0$ and $x^*_0$ seem to have the same meaning? If so, I suggest unifying the variable symbols.
4. As the paper mentioned, DPPS can be seen as a developed version of DPS. While I acknowledge that the proximal sampling strategy proposed in this paper is practical and novel, having only single novel method throughout the paper may not be sufficient to meet the novelty requirements expected by ACM MM as a top conference.

The 4th comment is the main reason why I tend to rate this paper as "Borderline". I feel that even if this work could add just one more contribution, I would be inclined to give it a higher rating.
For example, the revised work could attempt to make the posterior sampling strategy more intelligent, such as adaptively adopting different sampling frequencies for different timestamps to futher imporve image quality and reduce computational cost.

**Suitability:**

3

---

### Official Review · Reviewer_GFwW · 2024-05-28

**Rating:** 5
**Confidence:** 4

**Summary:**

The paper introduces a novel paradigm for diffusion-based image restoration called Diffusion Posterior Proximal Sampling (DPPS). It addresses the issue of over-smoothing in existing algorithms by selecting the most data-consistent sample from multiple candidates at each generative step, starting the restoration with an initialization that incorporates measurement signals. Extensive experiments demonstrate DPPS's superior performance in image restoration tasks with minimal additional computational cost.

**Strengths:**

1. **Improved Image Restoration Quality**: The proposed Diffusion Posterior Proximal Sampling (DPPS) method enhances image restoration performance by selecting the most data-consistent sample from multiple candidates, leading to more accurate and detailed results compared to existing methods.

2. **Efficient Use of Computational Resources**: DPPS achieves significant improvements in image restoration with only a marginal increase in computational cost, making it an efficient solution that does not require excessive computational resources.

3. **Robustness to Hyper-parameters**: The method demonstrates stability and robustness across a range of hyper-parameter configurations, meaning it can perform well without the need for fine-tuning to specific parameter settings.

**Limitations:**

The visual comparison images provided by the author are too few; additional visual comparison images for other tasks should be included, such as inpainting and deblurring. Moreover, the number of comparative methods is also too limited. It is suggested that the author compares more methods.

**Suitability:**

3

---

### Meta-Review · Area_Chair_HjQE · 2024-07-03

**Recommendation:** Accept (Oral)
**Confidence:** 5

**Metareview:**

This paper was reviewed by three experts in the field. The reviewers reached a consensus of positive feedback, which are Weak Accept, Weak Accept, and Accept. All reviewers appreciate the novelty of this proposed sampling strategy, which solves a long standing problem in the diffusion sampling process. Experimental results also show an impressive improvement. ACs also share the similar feedback as the reviewers. Based on this, the decision is to recommend the paper for acceptance to ACM Multimedia 2024.

We recommend that the authors carefully read all reviewers' final feedback, and revise the manuscript as needed. We congratulate the authors on the acceptance of their paper!